# *In vivo* study on the healing of bone defect treated with non-thermal atmospheric pressure gas discharge plasma

Akiyoshi Shimatani[1], Hiromitsu Toyoda[1]\*, Kumi Orita[1], Yoshihiro Hirakawa[2], Kodai Aoki[3], Jun-Seok Oh[3,4]\*, Tatsuru Shirafuji[3,4], Hiroaki Nakamura[1]

**1** Department of Orthopedic Surgery, Graduate School of Medicine, Osaka City University, Osaka, Japan, **2** Department of Orthopedic Surgery, Ishikiriseiki Hospital, Osaka, Japan, **3** Department of Physical Electronics and Informatics, Graduate School of Engineering, Osaka City University, Osaka, Japan, **4** BioMedical Engineering Center, Graduate School of Engineering, Osaka City University, Osaka, Japan

\* h-toyoda@msic.med.osaka-cu.ac.jp (HT); jsoh@osaka-cu.ac.jp (J-SO)

**Data Availability Statement:** All relevant data are within the paper.

**Funding:** The authors are highly grateful to JSPS KAKENHI (Grant Number JP19K03811), the Osaka

## Abstract

Medical treatment using non-thermal atmospheric pressure plasma (NTAPP) is rapidly gaining recognition. NTAPP is thought to be a new therapeutic method because it could generate highly reactive species in an ambient atmosphere which could be exposed to biological targets (e.g., cells and tissues). If plasma-generated reactive species could stimulate bone regeneration, NTAPP can provide a new treatment opportunity in regenerative medicine. Here, we investigated the impact of NTAPP on bone regeneration using a large bone defect in New Zealand White rabbits and a simple atmospheric pressure plasma (helium microplasma jet). We observed the recovery progress of the large bone defects by X-ray imaging over eight weeks after surgery. The X-ray results showed a clear difference in the occupancy of the new bone of the large bone defect among groups with different plasma treatment times, whereas the new bone occupancy was not substantial in the untreated control group. According to the results of micro-computed tomography analysis at eight weeks, the most successful bone regeneration was achieved using a plasma treatment time of 10 min, wherein the new bone volume was 1.51 times larger than that in the plasma untreated control group. Using H&E and Masson trichrome stains, nucleated cells were uniformly observed, and no inclusion was confirmed, respectively, in the groups of plasma treatment. We concluded the critical large bone defect were filled with new bone. Overall, these results suggest that NTAPP is promising for fracture treatment.

## Introduction

Bone fracture or loss is a common and serious medical problem that can result from various causes, including trauma, surgery, and degenerative diseases, and can significantly compromise a patient's quality of life. Reportedly, the incidence of fracture in those over 50 years of age is 116.5 per 10,000 people, and the risk of admission for fracture is 47.84 per 10,000 people [1,2]. In most cases, successful union is achieved by restoration of the alignment and stable

Medical Research Foundation for Intractable Diseases and the Nakatomi Foundation for their support. The funders had no role in study design, data collection and analysis, decision to publish, or preparation of the manuscript.

**Competing interests:** The authors have declared that no competing interests exist.

fixation of the fracture. However, roughly 5% of bone fractures fail to heal, resulting in non-union [3]. Patients with nonunion or segmental bone defects suffer long-term pain, physical disability, reduced quality of life, and significant treatment costs. These conditions are generally treated by operative means, including some form of bone fixation to provide adequate stability, decortication of the bone ends, and application of bone graft material to enhance the healing capacity [4]. Depending on the location and type of revision surgery, the success rate for nonunion ranges from 68% to 96% [5]. Therefore, treatment of delayed unions, nonunions, and bone loss poses a great challenge for orthopedic surgeons.

To date, numerous therapeutic approaches have been developed to enhance fracture healing capacity. Osteogenic bone formation mainly depends on different cell sources or specific cytokines, such as growth factors, and hormones. Therapeutic approaches for enhancing fracture healing can be classified as either biophysical or biological [6]. In the biophysical approach, electromagnetic fields and low-intensity pulsed ultrasonography are employed [7–11], whereas the biological approach comprises both local and systemic treatments strategies. Local biological strategies include the use of autologous bone marrow [12], peptide signaling molecules (fibroblast growth factor-2 and platelet-derived growth factors) [13,14], and morphogenetic factors (bone morphogenetic proteins and Wnt proteins) [14,15]. The systemic biological approach includes the use of parathyroid hormone [16], humanized monoclonal anti-sclerostin [17], or anti-Dickkopf-related protein 1 antibodies [18].

In recent years, the biomedical application of non-thermal atmospheric pressure plasma (NTAPP), hereafter referred to a plasma, has been gaining recognition [19]. It can be used at atmospheric pressure without considerably heating the background gas above ambient temperature. Plasma is well-known as the fourth state of matter, composed of partially ionized gas with electrons, ions, excited neutrals, and high energy photons. Those species do not exist in our daily environment, but interestingly are able to generate. Many studies have reported the biomedical applications of plasma, such as in wound healing, disinfection, and cancer treatment, in *in vitro* cell cultures [20–22]. As an indirect treatment, recent studies have established the potential advantages of plasma treatment in biomaterials used for bone and cartilage regeneration [23–25]. Moreover, it was reported that plasma can provoke differentiation and proliferation of stem cells that cause reactive oxygen species (ROS) generation [26]. Another study also suggested that plasma can induce osteogenic differentiation and enhance bone formation [27]. As our best knowledge, there is no report on the bone fracture therapy by direct treatment using NTAPP. The treatment of bone fracture and loss remains a challenging task, and advanced treatments or techniques for enhancing fracture healing capacity and repairing large bone defects are needed. In the present study, we focused on NTAPP gas discharge in addition to these approaches. Herein, we explored the effects of NTAPP on bone regeneration by using a single plasma treatment on a critical bone defect model in New Zealand White Rabbits.

## Materials and methods

### Helium microplasma jet treatment

The microplasma jet, hereafter termed plasma jet, used in this work was built in the laboratory and has been reported in previous studies (Fig 1A) [28–30]. Briefly, the plasma jet assembly is 150 mm long, with a 4 mm inner diameter and a 6 mm outer diameter glass tube that tapers to 650 μm at the nozzle. Power was supplied to a single 15 mm long external ring copper electrode wound onto the glass tube at a distance of 40 mm from the nozzle. The plasma jet assembly comprised a single electrode configuration. With this configuration, positive and negative discharges are alternatively generated. Thus, discharge bullet current(s) flow onto the biological target aka critical bone defect in this study.

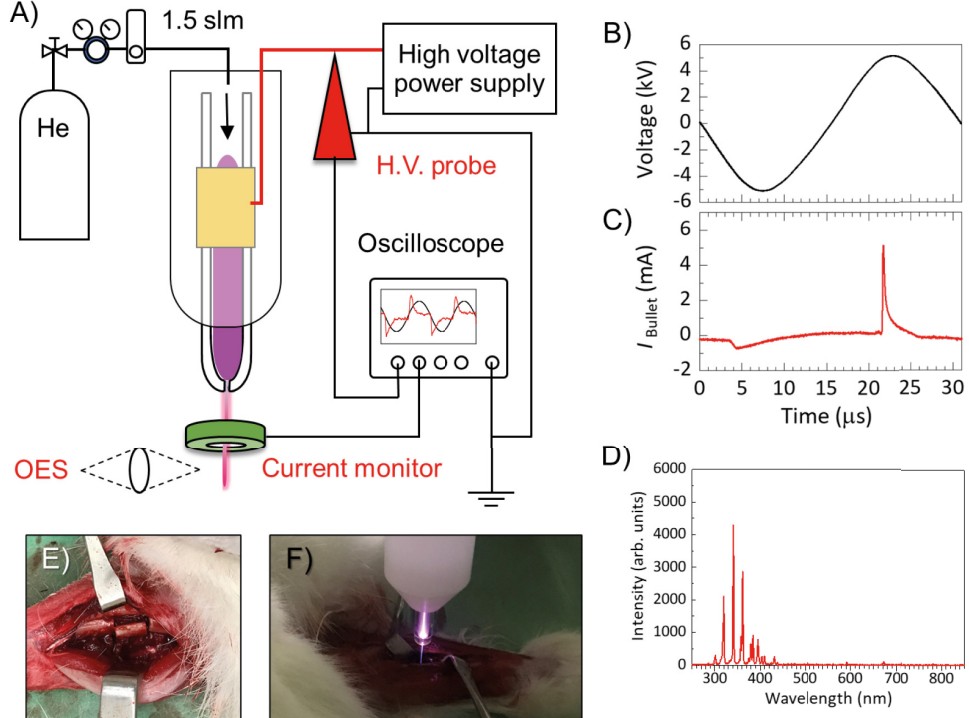

**Fig 1. Experimental set-up of helium microplasma jet to measure voltage and current measurements, and optical emission spectroscopy for understanding fundamental plasma properties.** 1B shows a sinusoidal high voltage waveform and typical bullet current. 1D shows an optical emission spectrum of plasma jet mainly $N_2$ 2nd positive systems. 1E shows a large bone defect model of ulna on the foreleg of New Zealand White Rabbits and 1F shows a plasma treatment on the large bone defect.

Helium (He) gas was used as a main discharge gas because it generates a stable glow discharge in an ambient air. Operating voltage is relatively lower than other gases (Ar, $N_2$ or air) and various ROS can be effectively generated by an interaction between highly reactive plasma species (electrons, $He^+$ and metastable $He^*$) and humid ambient air. The flow rate of He through the glass tube was fixed at 1.5 standard liters per minute. An optimized sinusoidal driving high voltage of 10 $kV_{p-p}$ (peak-to-peak) at 33 kHz was applied to the external electrode with a custom-made power supply (ORC Manufacturing Ltd., Japan) in cooperation. The voltage and jet current waveforms were measured by high voltage probe (PPE 20 kV, LeCroy, Chestnut Ridge, NY, USA) and a conventional current monitor (Pearson 2877, Pearson Electronics, Palo Alto, CA, USA), respectively, without a biological target due to a technical difficulty. The voltage and current waveforms were recorded with a digital oscilloscope (WaveJet 300A, LeCroy, Chestnut Ridge, NY, USA), as shown in Fig 1B and 1C. On a close inspection of Fig 1C, we see a typical plasma bullet current waveform: a wide and lower negative current pulse at 4 μs and a narrow and higher positive current pulse at 22 μs, respectively. An average input power of about 0.47 W was used to generate the outflow plasma jet, which was estimated by voltage and current measurements [28].

The optical emission spectrum of the plasma jet as shown in Fig 1D was measured by a fiber optic spectrometer (OceanOptics, Flame-TX-R1-ES, Largo, USA) associated with a collimating lens. The OES measurement was carried out on a free stream plasma jet. For plasma parameters, electron temperature ($T_e$) and gas temperature ($T_g$) were estimated by a spectrum simulator of the $N_2$ second positive system, which was kindly provided by professor Hiroshi

Akatsuka at the Tokyo Institute of Technology [29]. Using optical emission spectrum, $T_g$ of the plasma jet, 43.7˚C ± 5.3, was estimated by investigating the rotational temperature ($T_r$) of the $N_2$ second positive system ($C^3\Pi_u$ ($v' = 0,1$) and $B^3\Pi_g$ ($v'' = 2,3$) [30,31]. A 5-mm thick poly-tetrafluoroethylene housing was used to shield the high voltage electrode for safety. Under the parameters described above, the length of the free stream plasma jet, as seen with the unaided eye and measured with a ruler, was 12 mm.

## Subjects and surgery

Ten female New Zealand White rabbits, 13~16-weeks of age and weighing 3~3.5 kg, were obtained from SLC Japan Inc (Shizuoka, Japan). Rabbits were housed in a temperature-controlled environment (22˚C ± 2, humidity of 56% ± 5) with a cycle of 12 h of light and 12 h of dark throughout the experiment. All of the rabbits were fed ad libitum with a conventional balanced diet. Rabbits were anesthetized with ketamine hydrochloride, and a defect measuring 10 mm was made on both ulnar shafts using a bone saw (Fig 1E). The periosteum was resected with the bony segment 3–5 mm from the proximal and distal ends of the cut bony stumps. The periosteum of the adjacent radial surface was also removed, followed by irrigation to ensure maximum elimination of any periosteal tissue remnants. Rabbits were categorized into groups based on their plasma treatment time: 0 min (control group), 5 min, 10 min, and 15 min; each group was composed of five rabbits. Plasma was irradiated to the defect at a distance of 10 mm from the end of plasma jet. The plasma treatment was performed once at the time of surgery as shown in Fig 1F. Rabbits were weighed daily up to 3 days post-surgery and weekly thereafter and were monitored daily for potential signs of dehydration, pain, infection, and deviant behavior. After eight weeks, the rabbits were sacrificed by an overdose intravenous injection of pentobarbital sodium with little suffering, and the foreleg bones and tibia bones were removed for analysis. The surgery was performed by two authors (A.S and K.O). The animal protocol was approved by the Animal Ethics Committee of Osaka City University (permit number 15010 with date of approval 1 April 2018). All procedures were conducted in compliance with the Animal Research: Reporting of In Vivo Experiments guidelines.

## X-ray and micro-computed tomography analysis

X-rays of each foreleg bone were obtained every two weeks until eight weeks after surgery. X-ray images obtained eight weeks after the surgery were used to measure and compare the occupancy of the new bone with respect to the defect area. The defect area was defined as the area surrounded by the ulnar margin of the radius, straight lines extending the ulnar stump, and a straight line connecting the ulna side edges of the distal and proximal ulnar stumps. Measurement of the defect area and areas of new bone mass based on X-ray images was performed using ImageJ. To primarily evaluate osteogenesis of the radial cortex, the degree of radial cortex thickening was evaluated by a unique method. We divided the X-ray image into three categories according to the amount of cortex as follows: no evidence of cortex thickening, patchy cortex thickening, and cortical thickening exceeds 50% of the defect site length. Defect site length was defined as the length from the ulnar margin of radius to the straight line connecting the ulna side edges of the distal and proximal ulnar stumps. The length of cortical thickening was defined as the distance from the ulnar margin of the radius to the upper edge of the thickened cortical bone.

The extracted rabbit foreleg bones were fixed in a 10% neutral buffered formation solution at room temperature. Micro-computed tomography (μ-CT) was performed using SMX-90CT Plus, inspeXio (Shimadzu Corporation Japan). Scan data were reconstructed, and the volume of new bone was measured using three-dimensional image processing software (ExFact VR,

Nihon Visual Science, Inc. Japan). Image analysis was conducted by two authors (H.T. and Y. H.) in a blinded situation.

## Histological analysis

Because X-ray images cannot determine whether a bone defect is filled with new bone, tissue, or both, we performed a histological analysis for more detail of the recovery bone defect site. After the μ-CT analysis, each sample was decalcified with Morse solution (Wako Pure Chemical Industries, Ltd. Japan) and dehydrated using an alcohol series. Residual alcohol was removed by immersion in xylene and the tissue sample was embedded in paraffin block. Tissue slices (4 μm thick) were cut using a microtome and stained using hematoxylin and eosin (H&E) and Masson's trichrome stains, following standard protocols [32]. The sections were observed using a model BX53F microscope (Olympus, Japan) and photographed with an Olympus DP74 camera. Images were analyzed using Cellsens software (Olympus, Japan). Histological analysis was conducted by two authors (H.T. and Y.H.) in a blinded situation.

## XPS, surface wettability, thermal imaging

To identify surface chemistry and wettability changes, X-ray photoelectron spectroscopy (XPS) and the water contact angle (WCA) measurement were obtained. XPS analysis was performed on-site, with small pieces of tibia (around 7 mm × 7 mm). First, an untreated tibia sample was measured by XPS (ESCA-3400, Shimadzu, Japan). Then, the same tibia sample was removed and remeasured by XPS after being treated by the plasma jet for 10 min. Similarly, XPS analysis was performed for the artificial bone [β-tricalcium phosphate (β-TCP) SUPERPORE, HOYA technological CO., Japan]. In this experiment, 10 min of irradiation was selected to compare the group with the largest amount of new bone and the control group. Separately, WCA measurements were obtained via a contact angle analyzer (DMe-211, Kyowa Interface Sci. Co., Japan). A total of 1 μL of deionized water was dropped onto a section of the tibia, either untreated or plasma-treated. A plasma treatment time of 10 min was used for both XPS and WCA measurements. The thermal effect of the plasma treatment was investigated by a thermal imaging camera (T560, FLIR T560) and analyzed by the FLIR Tool$^+$ (FLIR®, USA). XPS, surface wettability, and thermal imaging analysis was conducted by three authors (K.A., J.-S.O., and T.S).

## Statistical analysis

Statistical analysis was performed using the Excel Statistics software for Windows (version 2019; SSRI Co. Ltd., Tokyo, Japan). Data are expressed as the means and standard deviations. Analysis of data was performed via a one-way analysis of variance, followed by a multiple comparison using the Tukey test. P values of <0.05 were considered statistically significant.

## Results

### X-ray imaging analysis

As seen in Fig 2A, in all cases, the defect site was filled in from the radius as a function of time. All defect sites that underwent plasma treatment tended to fill faster than the untreated controls. After eight weeks, average occupancy of the defect site in the10-min plasma-treatment group was more than 83%; 70% and 67% of the 15-min and 5-min plasma-treatment group were filled, respectively, and only 59% of the defect in the control group showed occupancy (Fig 2B). In addition, the highest occupancy score among each of the groups in order was as follows: 93% for the 10-min treatment group, 84% for the 5-min treatment group, 84% for the

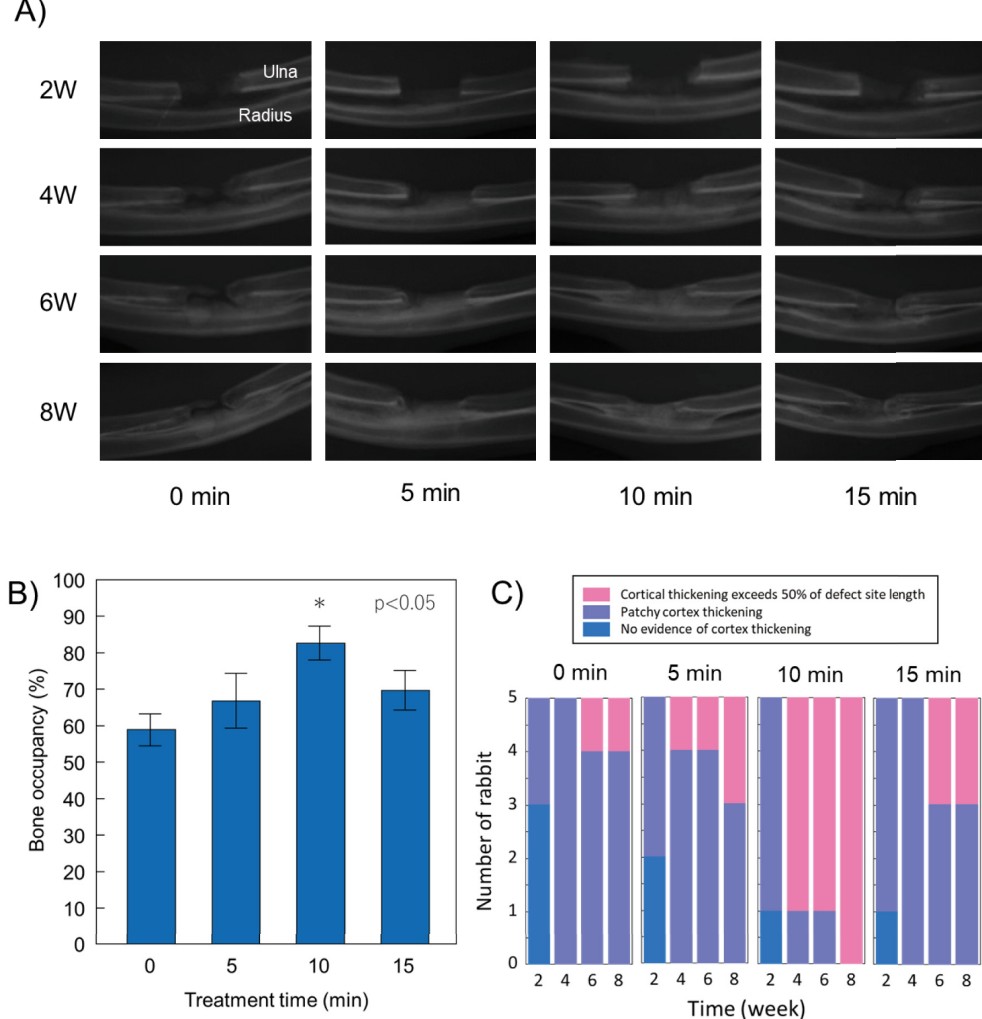

**Fig 2. X-ray imaging results of a large bone defect model of rabbit were taken up to 8 weeks.** 2A shows X-ray images as depended on the plasma treatment time. 2B shows bone occupancy of bone defect site at 8 weeks after a single plasma treatment and 2C shows evaluations of cortical thickening and patchy cortex thickening for treatment times at every two weeks.

15-min treatment group, and 67% for the control group. X-ray images of the control group at eight weeks after surgery revealed that the defect was only partially filled, and that filling was mainly composed of patchy cortex (Fig 2C). A close inspection of Fig 2C reveals no cortex thickening in 60% of the control cases, 40% in the 5-min treated cases, and 20% in the 10- and 15-min treated cases at week 2. Patchy cortex thickening and/or cortical thickening exceeding 50% of the defect site length clearly appeared in all cases at week 4. Interestingly, the radiological score was highest in the 10-min plasma-treated group, among whom 80% demonstrated cortical thickening exceeding 50% after week 4 (Fig 2C).

## Micro-computed tomography

After the rabbits were sacrificed at eight weeks, micro-computed tomography (μ-CT) of the defect was performed to estimate the volume of regenerated new bone aka fraction of bone occupancy, in the critical bone defect site. New bone mass was significantly higher in the

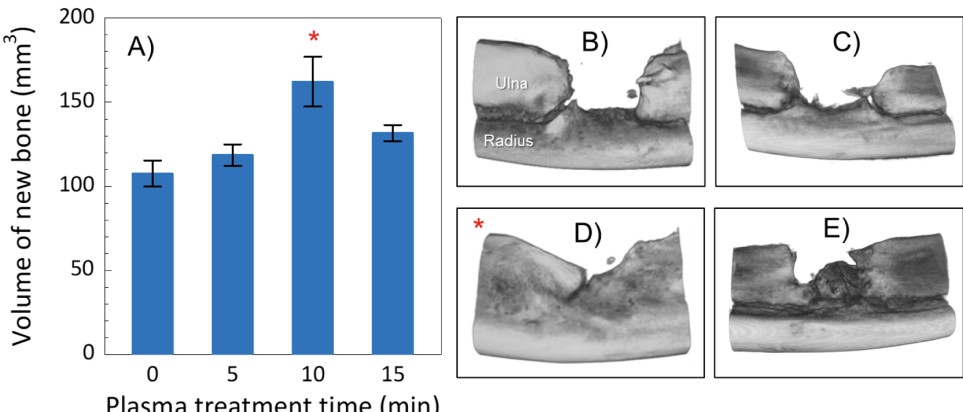

**Fig 3. Micro-computed tomography (μ-CT) analysis and pathology results.** 3A evaluated the volume of new bone as plasma treatment time and 3B-3E show μ-CT images of control (0 min) and plasma treatments: C for 5 min, D for 10 min, and E for 15 min, respectively.

group that underwent plasma treatment for 10 min compared with that of the control group (P = 0.0036) (Fig 3A). The average new bone volume of the 10 min plasma-treated group was 162.2 (± 14.8) mm$^3$, followed by 131.7 (± 4.8) mm$^3$ in the 15 min plasma-treated group; 118.6 (± 6.4) mm$^3$ in the 5 min; and 107.6 (± 7.8) mm$^3$ in the control group.

## Histological analysis

Representative images of sections stained with H&E and Masson trichrome stain are shown in Figs 4 and 5, respectively. Although some new bone was observed in the untreated control group (0 min), there were no cases in which the gap between the bone defects was continuously filled with new bone due to inclusions, such as fiber tissue and gaps, as expected. On the other hand, in the plasma irradiation groups, nucleated cells were uniformly observed in the H&E-stained image, and the Masson trichrome-stained image confirmed that there was no inclusion, and the bone defect was filled with new bone.

## Bone surface properties

The on-site XPS measurement shows a significant increase in O1s peak intensity (at 533 eV), but a decrease in C1s peak intensity (at 286 eV) after plasma treatment (Fig 6A). In addition, the basic components of bone (i.e., calcium and phosphorus) appeared after the 10-min plasma treatment, but not by the untreated bone. Unlike the artificial bone, muscles, ligaments, periosteum, etc. were attached to the surface of the living bone, and the peaks of calcium and phosphorus, which are pure bone components, observed in the living bone could be due to etching by plasma treatment. Related XPS peaks, Ca2s at 441 eV, Ca2p at 349 eV, P2s at 193 eV, and P2p at 135 eV, were clearly measured. It is well-known that the oxidized surface is strongly linked to the wet surface where cell adhesion is improved. We also confirmed a lower WCA of 42.0° ± 8.8 on the plasma-treated tibial surfaces compared to the WCA of 90.5° ±7.9 on untreated tibial surfaces.

Regarding the thermal effect (or damage) by plasma treatment, we found that the surface temperature was increased up to 43°C after the 10-min treatment. The tibial surface temperature was immediately increased from room temperature to around 40°C within 30 s, where it remained during the plasma treatment. The temperature suddenly decreased to nearly room temperature when the plasma jet was extinguished. These prompt temperature changes clearly

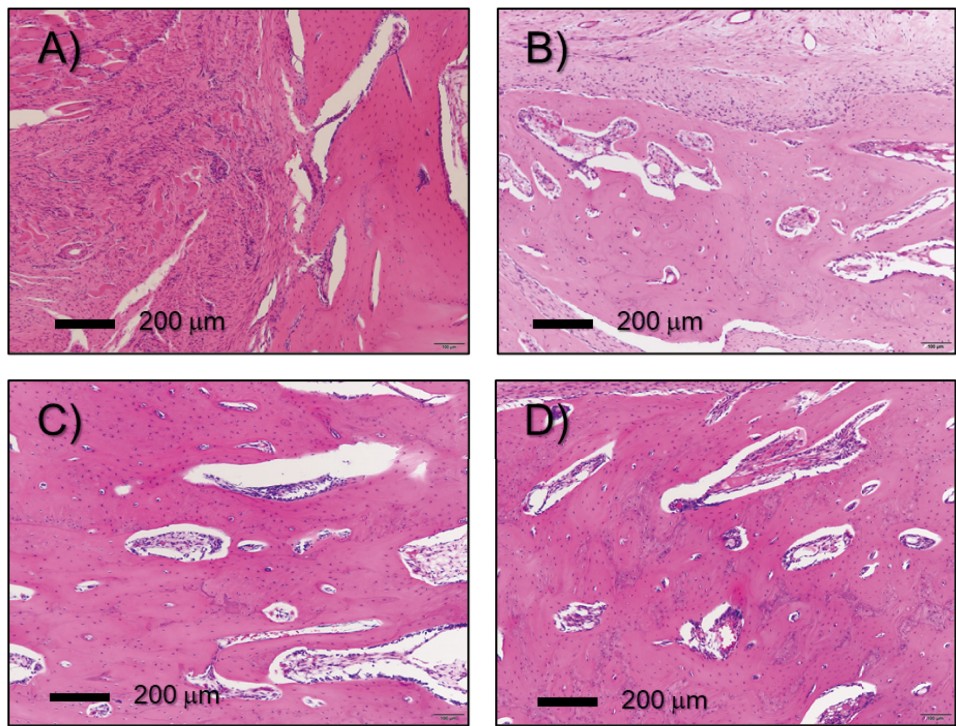

**Fig 4. Histological sections were stained with hematoxylin and eosin (H&E) as a function of plasma treatment time: 4A for 0 min, 4B for 5 min, 4C for 10 min, and 4D for 15 min, respectively.** Figures include a 200 μm scale bar.

illustrate that the temperature reflects the (plasma) gas temperature; furthermore, the depth of the temperature effect is shallow.

## Discussion

For the last decade, the potential use of NTAPP, with its lower temperature, has been explored in medical applications for the direct treatment of living tissues through sterilization, blood coagulation [33,34], wound healing, and tissue regeneration [35]. Furthermore, as an indirect treatment, recent studies have established the potential advantages of NTAPP treatment of bio-materials for use in bone and cartilage regeneration [23–25]. However, no previous studies have evaluated the direct bone regeneration promoting effect of NTAPP by irradiating a large bone defect site. Our survey is the first report to examine its impact on bone regeneration and to suggest a range of optimal treatment times by directly irradiating bone defects with NTAPP.

The balance between osteoblast-mediated bone formation and osteoclast-mediated bone resorption controls the bone remodeling process. In orthopedic surgery, sufficient bone regen-eration is especially needed to heal critical-sized bone defects after skeletal injury [35]. Meta-physeal defects of the long bone in animal models are commonly used to assess bone repair and regeneration because traumatic fractures often occur in the long bones, including the dis-tal radius, proximal femur and proximal humerus. Critical-sized bone defects are often created using animal models [36], of which rabbit ulnar or radial defects sized in a range between 10 and 20 mm have been widely used [37]. In particular, the rabbit ulnar bone was selected in this study because it is easily accessible, appropriately sized, easily assessed surgically as it is in a subcutaneous location with less soft tissue coverage, and has a splinting effect from the radius,

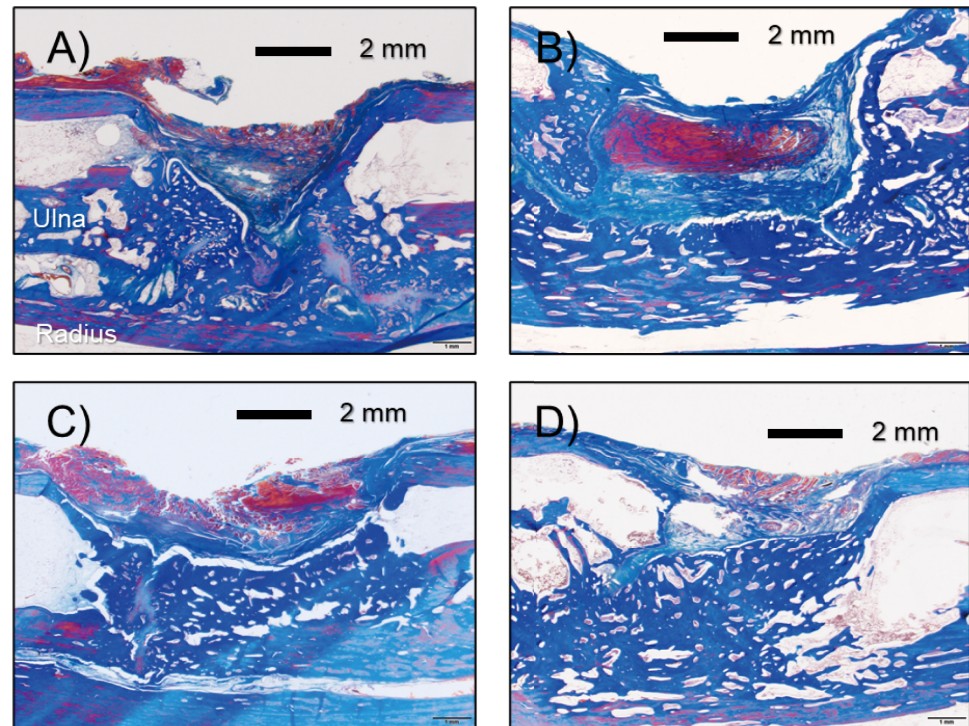

**Fig 5. Histological section was stained with Masson's trichrome as a function of plasma treatment time: 5A for 0 min, 5B for 5 min, 5C for 10 min, and 5D for 15 min, respectively.** Figures include a 2 mm scale bar.

so no implant is required. We created a critical bone defect so that there were no cases in which the defect merged in the control group. As a previous study reported, critical bone defects are usually unable to recover without any supporting material, such as artificial bone [38]. However, there were some cases in which the stumps of the ulna were fused to each other in the three plasma-treated groups, although the degree of fusion differed as we seen in Fig 2. Also, as mentioned above, the splinting effect was clearly seen in X-ray (Fig 2A), μ-CT (Fig 3), and histological section stained with Masson's trichrome (Fig 4). Especially, histological section stained with Masson's trichrome showed the bone defects was continuously filled with new bone. These results suggest that plasma treatment stimulated and promoted bone fusion. Based on the μ-CT results at eight weeks after the surgery, new bone mass was significantly higher in the groups that underwent plasma treatment for 10 min compared with the control group (P = 0.0036) (Fig 3). Although there was no statistically significant difference, the volume of new bone with 5 and 15 min treatments were higher than that in the control group (Fig 3A). This result suggests that an optimal treatment time exists, and, in this study, it was direct irradiation for about 10 min. In histological evaluation, the bone ingrowth of the plasma treated group was better than that of the non-treated control. Most of the new bones in the plasma group had continuity with the radial cortex, so it is possible that the cortical bone was significantly affected by the plasma treatment. So far, we understand that highly concentrated excited and ionized reactive oxygen and nitrogen species (RONS) are generated [39]. However, we do not know how the plasma was affected: which reactive species exactly stimulated and promoted new bone generation, and what amount of them was supplied to the bone defect site. Based on our best knowledge of the helium plasma jet and considering the situation of the direct treatment, highly reactive oxygen and nitrogen species (RONS) (e.g., hydroxyl radical

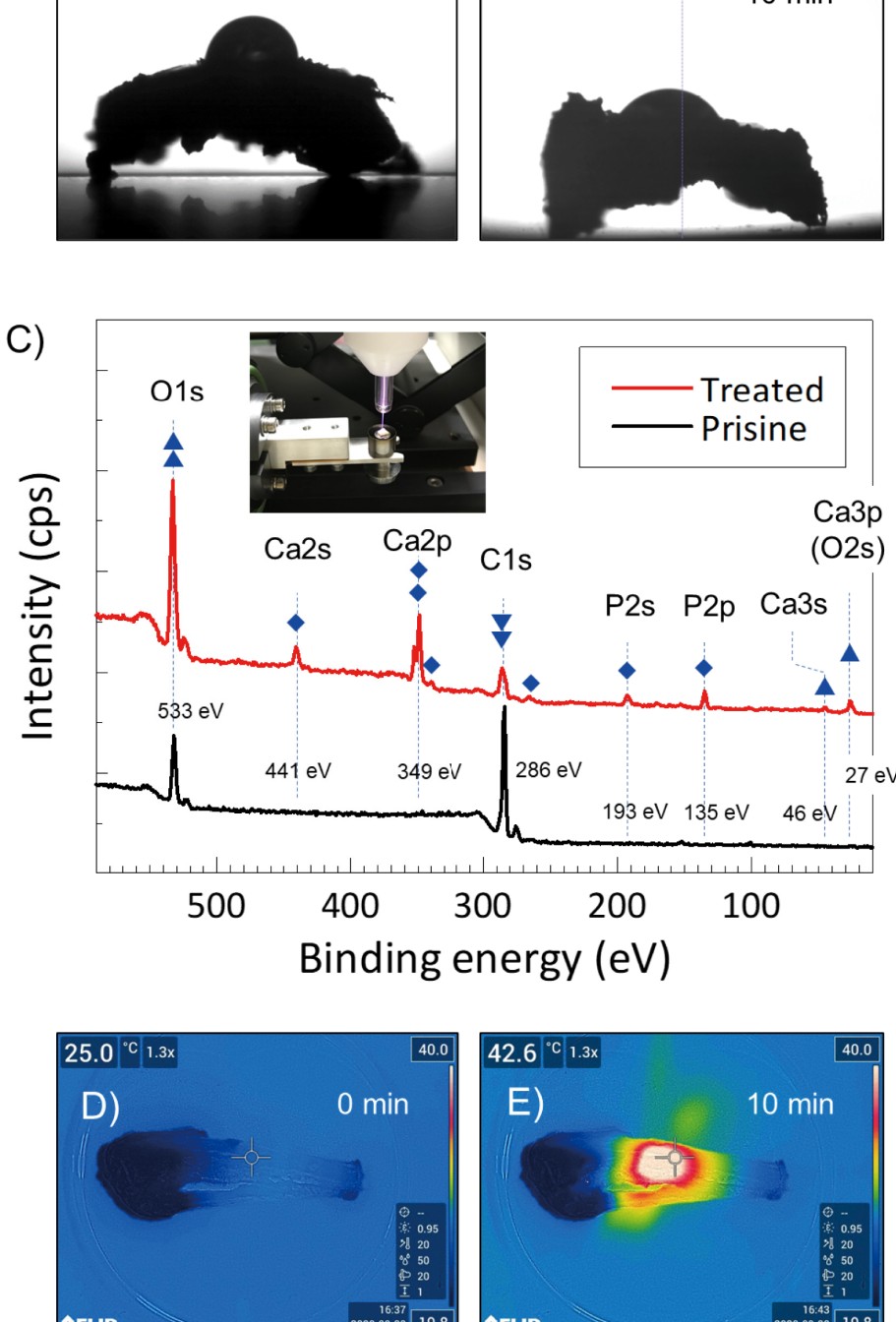

**Fig 6.** In comparison between 6A and 6B, wet surface after plasma treatment was observed. 6C shows a wide scan of XPS spectra measured on-site plasma treatment. Significant increase of O1s but decrease of C1s after the plasma treatment. Also, basic components of calcium and phosphorus are appeared after the treatment. Surface temperature increased around 43 °C after 10 min plasma treatment.

($^{\bullet}$OH), nitric oxide (NO), and oxygen-related neutral and positively and negatively charged species) could stimulate bone regeneration.

An *in vitro* study by Tominami *et al* reported enhanced osteoblast differentiation from atmospheric pressure plasma [40]. The study demonstrates the potential effect of the plasma treatment on bone regeneration and argued that short-lived but highly ROS (e.g., $^{\bullet}$OH and $^{\bullet}O_2^-$) are key components. In addition to being involved directly in cellular redox reactions, some of the RONS can also be dissolved in the liquid, culture medium, and cytoplasm to generate longer lived RONS e.g., $NO_2$, $NO_3$, ONOO, $O_2NOO$ [41].

In our study, since outflow plasma jet was a direct irradiated to the radius surface, it is possible that various ROS generated by an interaction between plasma and ambient air could affect bone regeneration as observed strong O1s peak on the plasma treated bone surface. In addition, these ROS can induce apoptosis at certain higher ROS concentrations [42]. It is considered that the two-sidedness of the plasma effect is one of the reasons an optimum irradiation time exists. Similar to phototherapy and radiotherapy, the biological effect of plasma depends on the "treatment dose" delivered into the targets (cell and living tissue, etc.). However, there have been no clear definition of plasma dose. A recent study proposed that the definition of plasma dose should be based on the dominant role of RONS in plasma biological effects [41]. Future research and clinical application can focus on in determining the appropriate plasma irradiation distance, time, etc. for future research and clinical application. In the early stages of osseointegration, the initial cell attachment, adhesion, proliferation, and differentiation of osteoblasts at the implant-bone interface play an important role. Chemical and physical properties of the biomaterial surface, such as wettability, roughness and topography, affect the behavior of the cells [43]. Surface wettability is one of the essential parameters affecting the biological response to biomaterials and implants. In general, it has been reported that high surface wettability promotes greater cell spreading and adhesion [44]. Plasma can modify the surface of materials economically and effectively by removing hydrocarbon and introducing the hydroxy group. These surface reactions enhance the surface wettability. Several studies have reported changes in surface biocompatibility in terms of cell attachment and protein adsorption with plasma treatment [45–47].

Although these reports are related to metals and biomaterials, in this study, strong generation of new bone was observed from the radial cortical bone, so we wondered if there is a similar change on the surface of the cortical bone. Thus, we compared the contact angles on the cortical bone surface of the tibia and found that the plasma-treated group was statistically and significantly more hydrophilic than the untreated control group (Fig 6A and 6B). As seen in Fig 6C, on-site XPS measurement showed a significant increase of O1 peak intensity after plasma treatment. Moderate oxidation of the bone surface by plasma irradiation is thought to be associated with increased hydrophilicity. We do not know the extent to which hydrophilic time-efficiency helps the attachment and growth of cells, as we did not identify any study that irradiated the bone surface during our study period. However, it may have a similar effect on the bone surface. We believe that research using cells is needed in our future study.

Regarding changes in surface temperature of bone due to plasma irradiation, we confirmed that 10 min of plasma treatment increased the surface temperature of tibia to around 43 $^{\circ}$C at maximum (Fig 6E). It has been reported that heat in the range of 40–45 $^{\circ}$C preserves bioactivity and is biocompatible with healthy bone cells [48]. Several previous studies have used osteoblast-like MC3T3 cells to analyze the thermal effects on bone formation *in vitro*. Daily thermal treatment for 10 minutes at 42$^{\circ}$C has been reported to increase the activity of alkaline phosphatase in MC3T3 cells in a time-dependent manner compared to non-thermal stress control [49]. It was also reported that 41$^{\circ}$C of heat shock causes upregulation of osteoblast differentiation from mesenchymal stem cells [50]. Furthermore, it has been reported that the induction of angiogenesis

by heating has the ability to stimulate new bone formation in and around the bone defect site [51]. From the above studies, it is considered that the increase in bone surface temperature due to the single event of plasma irradiation does not adversely affect bone formation. However, in this study, temperature change on the bone surface due to plasma irradiation was temporary and was unlikely to have a significant effect on the promotion of bone formation.

Now, we considering the plasma treatment on the cell proliferation and blood coagulation. An *in vitro* study reported that cold atmospheric plasma treatment of osteoblast-like (MG63) cells significantly upregulated Ki67 and PCNA, indicating active cell proliferation [52]. It was also reported that plasma can accelerate blood coagulation [53]. Hematomas form between the bone fragments at the bony injury site, wherein the stages of coagulation, inflammatory response, and healing take place. The early phase is characterized by high concentrations of mature granulocytes and monocytes/macrophages, as well as helper and cytotoxic lymphocytes in the hematoma. In addition, high levels of inflammatory and anti-inflammatory cytokines are found in the hematoma [54]. The cytokines present in the hematoma, as well as other growth factors, stimulate the differentiation of mesenchymal cells of the bone marrow toward the chondrogenic and osteogenic lines [55]. Results from the above reports, it is considered that promotion of cell proliferation and hematoma formation by plasma may promote bone regeneration.

As our best knowledge, this present study is the first demonstration that direct plasma treatment of bone defects increases new bone mass. Based on the results of the morphological and histological repair of bone defects, plasma exposure may effectively enhance fracture healing capacity and may, in the future, be used in clinics to shorten the bone union period and inhibit the occurrence of nonunion.

## Author Contributions

**Conceptualization:** Akiyoshi Shimatani, Hiromitsu Toyoda, Yoshihiro Hirakawa, Tatsuru Shirafuji, Hiroaki Nakamura.

**Data curation:** Akiyoshi Shimatani, Hiromitsu Toyoda, Kumi Orita, Yoshihiro Hirakawa, Kodai Aoki.

**Formal analysis:** Akiyoshi Shimatani, Kumi Orita, Yoshihiro Hirakawa, Kodai Aoki, Jun-Seok Oh.

**Funding acquisition:** Hiromitsu Toyoda, Tatsuru Shirafuji, Hiroaki Nakamura.

**Investigation:** Akiyoshi Shimatani, Hiromitsu Toyoda, Jun-Seok Oh, Tatsuru Shirafuji.

**Methodology:** Akiyoshi Shimatani, Hiromitsu Toyoda, Hiroaki Nakamura.

**Project administration:** Hiromitsu Toyoda, Tatsuru Shirafuji, Hiroaki Nakamura.

**Supervision:** Hiromitsu Toyoda, Jun-Seok Oh, Tatsuru Shirafuji, Hiroaki Nakamura.

**Validation:** Akiyoshi Shimatani, Hiromitsu Toyoda, Kumi Orita, Kodai Aoki, Jun-Seok Oh.

**Visualization:** Akiyoshi Shimatani, Kodai Aoki, Jun-Seok Oh.

**Writing – original draft:** Akiyoshi Shimatani, Hiromitsu Toyoda, Kumi Orita, Jun-Seok Oh.

**Writing – review & editing:** Akiyoshi Shimatani, Hiromitsu Toyoda, Yoshihiro Hirakawa, Jun-Seok Oh, Tatsuru Shirafuji, Hiroaki Nakamura.

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
