## [Decision Letter · Decision Letter 0]

17 Aug 2021

PONE-D-21-22730

In vivo study on the healing of bone defect treated with non-thermal atmospheric pressure gas discharge plasma

PLOS ONE

Dear Dr. Oh,

Thank you for submitting your manuscript to PLOS ONE. After careful consideration, we feel that it has merit but does not fully meet PLOS ONE’s publication criteria as it currently stands. Therefore, we invite you to submit a revised version of the manuscript that addresses the points raised during the review process.

We look forward to receiving your revised manuscript.

Kind regards,

Nagendra Kumar Kaushik, PhD

Academic Editor

PLOS ONE

Journal Requirements:

 [K.O., H.T., J.-S.O, and T.S. acknowledge a JSPS KAKENHI Grant Number JP19K03811. H.T. thanks financial support from  the Osaka Medical Research Foundation for Intractable Diseases and the Nakatomi Foundation. J.-S.O and T.S. thank for financial support from the Osaka City University (OCU) Strategic Research Grant 2019 for Top Priority Researches.]

[K.O., H.T., J.-S.O, and T.S. acknowledge a JSPS KAKENHI Grant Number JP19K03811. H.T. thanks financial support from  the Osaka Medical Research Foundation for Intractable Diseases and the Nakatomi Foundation. J.-S.O and T.S. thank for financial support from the Osaka City University (OCU) Strategic Research Grant 2019 for Top Priority Researches.]

 [K.O., H.T., J.-S.O, and T.S. acknowledge a JSPS KAKENHI Grant Number JP19K03811. H.T. thanks financial support from  the Osaka Medical Research Foundation for Intractable Diseases and the Nakatomi Foundation. J.-S.O and T.S. thank for financial support from the Osaka City University (OCU) Strategic Research Grant 2019 for Top Priority Researches.]

5. Please ensure that you refer to Figure 5 in your text as, if accepted, production will need this reference to link the reader to the figure.

Additional Editor Comments:

I recommend revising this manuscript as per both reviewer comments.

Reviewers' comments:

Reviewer's Responses to Questions

**Comments to the Author**

1. Is the manuscript technically sound, and do the data support the conclusions?

Reviewer #1: Yes

Reviewer #2: Yes

2. Has the statistical analysis been performed appropriately and rigorously? 

Reviewer #1: Yes

Reviewer #2: Yes

3. Have the authors made all data underlying the findings in their manuscript fully available?

Reviewer #1: Yes

Reviewer #2: Yes

4. Is the manuscript presented in an intelligible fashion and written in standard English?

Reviewer #1: Yes

Reviewer #2: Yes

5. Review Comments to the Author

Reviewer #1: The submitted paper Shimatani et. all explain the use of plasmas for biomedical applications in encountering a growing interest, especially in the framework of In vivo study on the healing of bone defects, which aims at exploiting the action of low-power, atmospheric pressure plasmas for therapeutic purposes. Several applications have already reached the stage of clinical trials, while others are on their way, a large set of different plasma sources able to work at atmospheric pressure with low dissipated power has been created, and some of them are already certified as medical devices. From the scientific viewpoint, Overall, the manuscript reports on an interesting and well-written. It is suitable for publication after clarification on the points raised below:

(1) Please try to make uniforms, like "non-thermal atmospheric pressure plasma (NTAPP)", "non-thermal atmospheric pressure gas discharge plasma", and "gas discharge plasma". Select uniform terminology.

(2) Overview of recent progress in applications of non-thermal plasma for therapeutic approaches has been developed to bone fracture healing and regeneration is need to be extended in the introduction.

(3) Brief discussion about various RONS in the term of oxidation potential is needed in the introduction [ I refer https://doi.org/10.1016/j.cpme.2017.12.052, and https://doi.org/10.1016/j.cpme.2017.12.052 paper for a better explanation of plasma chemistry.]

(4) Regarding the experimental setup, the authors claimed that, "The plasma jet assembly comprised a single electrode configuration. Thus, discharge currents passed through the biological target underneath the plasma." If so then why don't you select DC power-driven plasma source? The application of AC-driven plasma sources doesn't support your statement. Please supply some reasons/advantages of using AC have driven power source. For in vivo study, the floating jet is not electrically safe and it requires a much more breakdown voltage as compared to two electrode plasma jets. I think the logical explanation about why floating plasma jet is used in this experiment should be added.

(5) This effect seen on bone is due to RONS or discharge current? Do you perform an experiment to support "discharge currents passed through the biological target underneath the plasma that will show a better bone healing effect"? If this statement is based on literature please cite the appropriate reference. Comparative study of dc- and ac-driven plasma source could support this statement.

(6) Current-voltage waveform and OES are greatly affected by target and without a target in this type of floating jet. Figure 1B, C, and D were measured with the target or without a target? Besides, please supply some reasons/advantages of using AC has driven power source.

(7) Author assumed that highly reactive oxygen and nitrogen species (RONS) (e.g., hydroxyl radical OH), nitric oxide (NO), and oxygen-related neutral and positively and negatively charged species)could stimulate bone regeneration, for this why author selected helium gas? Can we expect a similar result in other economic working gas like argon, nitrogen, and air?

(8) why there is only one positive current peak?

At last, the Figures in the submitted manuscript are inverted. Please consider this during the submission of a revised manuscript.

Reviewer #2: This manuscript (PONE-D-21-22730) entitled “In vivo study on the healing of bone defect treated with non-thermal atmospheric pressure gas discharge plasma” by Oh et al. showed that non-thermal atmospheric pressure gas discharge plasma is promising for fracture treatment.

They investigated the impact of plasma on bone regeneration using a large bone defect in model rabbits and simple helium microplasma jet operated at atmospheric pressure. Overall structure of manuscript is acceptable but the writing should be revised thoroughly to make it understand better. Despite the concept of the newly therapeutic strategy for bone repair, this article is still not acceptable to have publication in PloS ONE journal with current form. Some specified minor comments are highlighted as below which should be carefully considered and revised accordingly prior to acceptance.

1. Abstract should be more effective and precise with only main findings of the manuscript wi.

2. Authors should add some in vitro data to confirm the specificity of plasma treatment on bone cell proliferation.

3. Did the authors tested their helium plasma jet effect on physiological solution. Author should refer and cite this article regarding this (Pulsed 3.5 GHz high power microwaves irradiation on physiological solution and their biological evaluation on human cell lines. Sci Rep 11, 8475, 2021. Brief experiments and explanations are required.

6. PLOS authors have the option to publish the peer review history of their article (what does this mean?). If published, this will include your full peer review and any attached files.

Reviewer #1: No

Reviewer #2: No

---

## [Author Response · Author response to Decision Letter 0]

20 Sep 2021

Dear reviewers 

Thank you very much for your valuable comments and questions. 

We reply to your comments and question in this revision. 

Please see below and please find an attached Response to Reviewers. 

Yours, 

Jun-Seok Oh

Response to Reviewers (PONE-D-21-22730)

Reviewer #1: The submitted paper Shimatani et. all explain the use of plasmas for biomedical applications in encountering a growing interest, especially in the framework of In vivo study on the healing of bone defects, which aims at exploiting the action of low-power, atmospheric pressure plasmas for therapeutic purposes. Several applications have already reached the stage of clinical trials, while others are on their way, a large set of different plasma sources able to work at atmospheric pressure with low dissipated power has been created, and some of them are already certified as medical devices. From the scientific viewpoint, Overall, the manuscript reports on an interesting and well-written. It is suitable for publication after clarification on the points raised below:

Reviewer #1’s comment 1 Please try to make uniforms, like "non-thermal atmospheric pressure plasma (NTAPP)", "non-thermal atmospheric pressure gas discharge plasma", and "gas discharge plasma". Select uniform terminology. 　　　

Our reply for reviewer #1’s comment 1, 

Thank you for comment, we prefer to use ‘non-thermal atmospheric pressure plasma (NTAPP)’ through the manuscript. We amended through the manuscript.

Reviewer #1’s comment 2 Overview of recent progress in applications of non-thermal plasma for therapeutic approaches has been developed to bone fracture healing and regeneration is need to be extended in the introduction. 

Our reply for reviewer #1’s comment 2, 

As our best knowledge, there are no reports related on the bone fracture healing and regeneration. All related studies are focused on the plasma treatment for materials such as an implantation. We discussed these issues in the later part of submitted manuscript. 

Reviewer #1’s comment 3 Brief discussion about various RONS in the term of oxidation potential is needed in the introduction [ I refer https://doi.org/10.1016/j.cpme.2017.12.052, and https://doi.org/10.1016/j.cpme.2017.12.052 paper for a better explanation of plasma chemistry.] 

Our reply for reviewer #1’s comment 3, 

Thank for the comment. But unfortunately, we couldn’t find enough information various RONS in the term of oxidation potential in the suggested article. 

Reviewer #1’s comment 4 Regarding the experimental setup, the authors claimed that, "The plasma jet assembly comprised a single electrode configuration. Thus, discharge currents passed through the biological target underneath the plasma." If so then why don't you select DC power-driven plasma source? The application of AC-driven plasma sources doesn't support your statement. Please supply some reasons/advantages of using AC have driven power source. For in vivo study, the floating jet is not electrically safe and it requires a much more breakdown voltage as compared to two electrode plasma jets. I think the logical explanation about why floating plasma jet is used in this experiment should be added. 

Our reply for reviewer #1’s comment 4, 

Thank you for the comment. We amended the sentences below.

‘The plasma jet assembly comprised a single electrode configuration. With this configuration, positive and negative discharges are alternatively generated and following discharge bullet current(s) are flown onto the biological target aka critical bone defect in this study.’ 

Reviewer #1’s comment 5 This effect seen on bone is due to RONS or discharge current? Do you perform an experiment to support "discharge currents passed through the biological target underneath the plasma that will show a better bone healing effect"? If this statement is based on literature please cite the appropriate reference. Comparative study of dc- and ac-driven plasma source could support this statement. 

Our reply for reviewer #1’s comment 5, 

We do not fully understand the mechanism and we still investigating how plasma can stimulate on the bone regeneration. It seems we need to separate the two parameters, RONS and currents, in our future study. 

Reviewer #1’s comment 6 Current-voltage waveform and OES are greatly affected by target and without a target in this type of floating jet. Figure 1B, C, and D were measured with the target or without a target? Besides, please supply some reasons/advantages of using AC has driven power source. 

Our reply for reviewer #1’s comment 6, 

Thank you for the comment. Current-voltage waveform and OES data shown in the manuscript for the freestream jet. In case of bullet current measurement for the freestream plasma jet using current monitor is technically difficult. Because we have similar plume length and thickness of current monitor around 15 mm. For OES measurement, we carried out both cases with and without bone. But we have no significant difference for both cases, unlikely wet or liquid targets. 

We amended related parts in the manuscript below. 

‘The voltage and jet current waveforms were measured by high voltage probe (PPE 20 kV, LeCroy, Chestnut Ridge, NY, USA) and a conventional current monitor (Pearson 2877, Pearson Electronics, Palo Alto, CA, USA), respectively, without a biological target due to a technical difficulty.’

‘The optical emission spectrum of the plasma jet as shown in Fig 1D was measured by a fiber optic spectrometer (OceanOptics, Flame-TX-R1-ES, Largo, USA) associated with a collimating lens. The OES measurement was carried out on a free stream plasma jet.’

Reviewer #1’s comment 7 Author assumed that highly reactive oxygen and nitrogen species (RONS) (e.g., hydroxyl radical OH), nitric oxide (NO), and oxygen-related neutral and positively and negatively charged species)could stimulate bone regeneration, for this why author selected helium gas? Can we expect a similar result in other economic working gas like argon, nitrogen, and air? 

Our reply for reviewer #1’s comment 7, 

Thank you for your question. Our answer is here. Helium gas can effectively generate various RONS by an interaction between plasma species (electrons, positive helium ions and especially metastable helium) and humid ambient air. And the stable (laminar) gas flow support to carry highly reactive but short living species on the target without any addition reaction. And we accept reviewer’s comment helium gas is expensive. But discharge with other gases e.g., argon, nitrogen, and air, are usually unstable and nonuniform streamer, while helium to be a glow discharge in atmospheric pressure. This is why we prefer to use helium gas. 

We added sentences ‘Helium (He) gas was used as a main discharge gas because it generates a stable glow discharge in an ambient air. Operating voltage is relatively lower than other gases (Ar, N2 or air) and various ROS can be effectively generated by an interaction between highly reactive plasma species (electrons, He+ and metastable He*) and humid ambient air.’ 

Reviewer #1’s comment 8 why there is only one positive current peak?

At last, the Figures in the submitted manuscript are inverted. Please consider this during the submission of a revised manuscript. 

Our reply for reviewer #1’s comment 8, 

Thank you for the question. 

If we see the figure again, reviewer can see the negative current pulse, which is a small and broad. We added a sentence ‘On a close inspection of Fig 1C, we see a typical plasma bullet current waveform: a wide and lower negative current pulse at 4 µs and a narrow and higher positive current pulse at 22 µs, respectively.’ 

Reviewer #2: This manuscript (PONE-D-21-22730) entitled “In vivo study on the healing of bone defect treated with non-thermal atmospheric pressure gas discharge plasma” by Oh et al. showed that non-thermal atmospheric pressure gas discharge plasma is promising for fracture treatment.

They investigated the impact of plasma on bone regeneration using a large bone defect in model rabbits and simple helium microplasma jet operated at atmospheric pressure. Overall structure of manuscript is acceptable but the writing should be revised thoroughly to make it understand better. Despite the concept of the newly therapeutic strategy for bone repair, this article is still not acceptable to have publication in PloS ONE journal with current form. Some specified minor comments are highlighted as below which should be carefully considered and revised accordingly prior to acceptance.

Reviewer #2’s comment 1 Abstract should be more effective and precise with only main findings of the manuscript wi. 

Our reply for reviewer #2’s comment 1, 

Thank you for the comment. We amended the abstract. 

‘Medical treatment using non-thermal atmospheric pressure plasma (NTAPP) is rapidly gaining recognition. NTAPP is thought to be a new therapeutic method because it could generate highly reactive species in an ambient atmosphere which could be exposed to biological targets (e.g., cells and tissues). If plasma-generated reactive species could stimulate bone regeneration, NTAPP can provide a new treatment opportunity in regenerative medicine. Here, we investigated the impact of NTAPP on bone regeneration using a large bone defect in New Zealand White rabbits and a simple atmospheric pressure plasma (helium microplasma jet). We observed the recovery progress of the large bone defects by X-ray imaging over eight weeks after surgery. The X-ray results showed a clear difference in the occupancy of the new bone of the large bone defect among groups with different plasma treatment times, whereas the new bone occupancy was not substantial in the untreated control group. According to the results of micro-computed tomography analysis at eight weeks, the most successful bone regeneration was achieved using a plasma treatment time of 10 min, wherein the new bone volume was 1.51 times larger than that in the plasma untreated control group. Using H&E and Masson trichrome stains, nucleated cells were uniformly observed, and no inclusion was confirmed, respectively, in cases of plasma treatment. We concluded the critical large bone defect were filled with new bone. Overall, these results suggest that NTAPP is promising for fracture treatment.’

Reviewer #2’s comment 2 Authors should add some in vitro data to confirm the specificity of plasma treatment on bone cell proliferation. 

Our reply for reviewer #2’s comment 2, 

Thank you for the comment. 

Here we focused on in vivo and a related report in a reference 27 was introduced i.e., plasma can induce osteogenic differentiation and enhance bone formation. Some in vitro studies, references 27, 45, 46, and 52, have already indicated that NTAPP enhance the initial cell attachment, proliferation, and differentiation of osteoblasts. We have already cited these reports in the Discussion section. Thus, we feel it is unnecessary to add in vitro data in this study. If necessary, it should report, separately, because the mechanisms are different: RONS in gas phase stimulate biological reaction in case of in vivo, while RONS in liquid phase do in case of in vitro. 

Reviewer #2’s comment 3 Did the authors tested their helium plasma jet effect on physiological solution. Author should refer and cite this article regarding this (Pulsed 3.5 GHz high power microwaves irradiation on physiological solution and their biological evaluation on human cell lines. Sci Rep 11, 8475, 2021. Brief experiments and explanations are required. 

Our reply for reviewer #2’s comment 3, 

Thank you for the question. 

Unfortunately, we have no chance to test the helium plasma jet effect on physiological solution. Also, we considered the suggested reference, but we cannot find right place in our manuscript. Sorry to this.

---

## [Editor Report · Decision Letter 1]

22 Sep 2021

In vivo study on the healing of bone defect treated with non-thermal atmospheric pressure gas discharge plasma

PONE-D-21-22730R1

Dear Dr. Oh,

We’re pleased to inform you that your manuscript has been judged scientifically suitable for publication and will be formally accepted for publication once it meets all outstanding technical requirements.

Kind regards,

Nagendra Kumar Kaushik, PhD

Academic Editor

PLOS ONE

Additional Editor Comments (optional):

No comments
---

## [Editor Report · Acceptance letter]

29 Sep 2021

PONE-D-21-22730R1 

*In vivo* study on the healing of bone defect treated with non-thermal atmospheric pressure gas discharge plasma 

Dear Dr. Oh:

I'm pleased to inform you that your manuscript has been deemed suitable for publication in PLOS ONE. Congratulations! Your manuscript is now with our production department. 

Kind regards, 

on behalf of

Prof. Nagendra Kumar Kaushik 

Academic Editor

PLOS ONE